# Pooled prevalence and associated factors of diarrhea among under-five years children in East Africa: A multilevel logistic regression analysis

**Abiyu Abadi Tareke**[1]*, **Ermias Bekele Enyew**[2], **Bayley Adane Takele**[3]

**1** Department of Monitoring and Evaluation, West Armachiho District Health Office, Gondar, Ethiopia, **2** Department of Health Informatics, Mettu University, Mettu, Ethiopia, **3** Department of Medical Auditing, Ethiopian Health Insurance Agency Gondar Branch, Gondar, Ethiopia

* abiyu20010@gmail.com

## Abstract

**Data Availability Statement:** The data underlying the results presented in the study are available from www.dhsprogram.com.

### Background

Worldwide, diarrhea is the second most common cause of death and morbidity among under -five years' children. In sub-saran Africa, access to water, sanitation, and hygiene are very scanty and the burden of diarrhea diseases is countless relative to the rest of the world. Prior studies conducted in East Africa vary in design, sample size, and other data collection tools. Through those studies, it is hard to make regional comparisons. Combining datasets that are studied on similar people and having common variable identified enhances statistical power due to the large sample size, advance the ability to compare outcomes, and create the opportunity to develop new indicators. Hence, this study aimed to assess the prevalence and associated factors of diarrhea among under five years' children using the most recent national representative Demographic and Health Surveys from 12 East African countries. The information generated from this pooled datasets will give good insight into the sub-regional prevalence of diarrhea.

### Methods

This study utilized secondary data from 12 East African countries' most recent demographic health survey. Variables were extracted and appended together to assess the pooled prevalence of diarrhea and associated factors. A total of 90,263 under-five years of age children were encompassed in this study. STATA version was used to cross-tabulate and fit the models. To account for the hierarchical nature of the demographic health survey, multilevel logistic regression was calibrated. BIC, AIC, deviance, and LLR were used as Model comparison parameters. Variables with a p-value of <0.2 were considered for multivariable analysis. Adjusted odds ratio with 95% CI and p-value <0.05 were used to declare statistical significances of factors.

**Funding:** The authors received no specific funding for this work.

**Competing interests:** The authors have declared that no competing interests exist.

**Abbreviations:** AIC, Akaike information criteria; AOR, Adjusted Odds Ratio; BIC, Bayesian information criteria; BF, Breast Feeding; DHS, Demographic health survey; DIC, Deviance Information Criterion; ICC, Intra-Class Correlation; LLR, Log- Likelihood Ratio; MOR, Median Odds Ratio and; PCV, Proportion of Variance Change.

## Results

The pooled prevalence of diarrhea in under five years children was 14.28% [95%CI; 14.06%, 14.51%]. Being child whose mother age is 15–24 years [AOR = 1.41, 95% CI; 1.33, 1.49], 25–34 years[AOR = 1.17, 95%CI; 1.10, 1.23], being 7–12 months child [AOR = 3.10, 95%CI; 2.86, 3.35], being 12–24 months child [AOR = 2.56, 95%CI; 2.38, 3.75], being 25–59 months child [AOR = 0.88, 95%CI; 0.82, 0.95], being child from poor household [AOR = 1.16, 95%CI; 1.09, 1.23], delayed breast feeding initiation (initiated after an hour of birth) [AOR = 1.15, 95%CI; 1.10, 1.20], and being a child from community with low educational status [AOR = 1.10, 95%CI; 1.03, 1.18] were factors associated with diarrheal diseases.

## Conclusion

The pooled prevalence of diarrhea among under five years of children in East African countries is high. Maternal age, child's age, wealth status of the household, the timing of breast feeding initiation, sex of the child, community level of educational status, working status of the mother, and the number of under five children were factors that were associated with diarrheal diseases. Scaling up of maternal and child health services by government and other concerned bodies should consider those economically marginalized communities. Additionally, awareness should be created for those uneducated mothers concerning the nature of childhood diarrhea.

## Background

Diarrhea in children is defined as, three or more bowel movements (passage of loose stool) per 24 hours or watery stool that is different from normal [1]. Worldwide, diarrhea is the second cause of death and morbidity among under five years children. Each year 525, 000 under five years children lost their lives due to avoidable diarrhea diseases, and Globally, there are approximately 1.7 billion cases of childhood diarrheal case every year [2]. According to the report of Global Burden of Diseases(GBD) in 2019, in sub-Saharan Africa total Disability-adjusted life years (DALYs) estimate due to diarrhea was 13.01% [3, 4]. As poor access to sanitary materials is the main case of diarrheal diseases [5], In sub-saran Africa, access to water, sanitation, and hygiene (WASH) are very scanty [5] and the burden of diarrhea diseases is countless relative to the rest of the world. As claimed by the Global Burden of Diseases visual hub, total DALYS among under five years children in the Eastern sub-saran region contributed by diarrheal diseases was 10.21% [4]. Different studies showed that the prevalence of diarrheal diseases among children under five years is high in East African countries. Based on meta-analysis conducted in Ethiopia the prevalence of diarrhea ranges from 19% to 25% [6]. Other studies conducted in Uganda, Rwanda, and Malawi uncovered that the prevalence of diarrheal diseases was 32% [7], 26.7% [8], and 20% [9] respectively. Additionally, the culture of open defecation in East African countries is common. For example, a systematic review done in Ethiopia showed a low level of open defecation free areas(i.e. 16%) [10]. Other studies also showed a low level of open defecation free communities like Kenya (14%) [11], Uganda (22.9%) [12]. In general, in East African countries, access to basic sanitation and hygiene services is very low [13].

Prior studies disclosed that many factors are associated with diarrhea among under five years children. Among those; child's age [7, 14–16], mother/caregiver's educational status [8, 14–18], place of residency [7, 15, 19], not vaccinated to Rota [8], maternal/caregiver's age [17], travel time to the water sources [17], wealth index [7, 15, 17, 18], birth interval [16], unimproved sources of drinking water [16], exclusive breast feeding, timing of breast feeding initiation [7], regional location [7], mother's working status [7] and unimproved toilet facilities [9].

So far, many studies have been done in East Africa about diarrheal diseases among under five years children. However, those individual studies vary in design and sample size, which is difficult to perform the regional comparison. Making a regional comparison is important to meet current global initiatives agendas like Sustainable Development Goal (SDG). Combining datasets that are studied on similar people and having common variable identifiers enhances statistical power due to the large sample size, advances the ability to compare outcomes and paves the way to develop new indicators. To date, in East Africa, studies conducted to describe diarrheal diseases among under five years children by merging cross-national datasets are limited. Hence, this study aimed to assess the prevalence and associated factors of diarrhea among under five years children using the most recent (2008–2019) nationally representative Demographic and Health Surveys (DHS) from 12 East African countries. The Information generated from this pooled data will give a good insight into the sub-regional prevalence of diarrhea. This study might also help policy makers, global organizations, NGOs, and researchers to identify the most vulnerable East African region to diarrhea, to give urgent interventional measures and resource allocation. This study found that the conjoined prevalence of diarrheal diseases in east Africa high and modifiable factors like wealth status of households, time of breast feeding initiation and educational status were the main determinants of diarrheal episodes among under-five children.

## Methods and materials

### Data sources

As the majority of the population of East African countries are rural residents, more than fifty percent of the resident of East African countries lacks improved WASH indicators [20]. East African countries are countries with the highest prevalence of diarrheal diseases among under-five children when compared to the rest of the world [17]. This study used data from 12 Eastern African countries of most up to dated demographic health surveys. Eastern African countries embodied in this study were Burundi, Ethiopia, Comoros, Uganda, Rwanda, Tanzania, Mozambique, Madagascar, Zimbabwe, Kenya, Zambia, and Malawi. Mayotte, Reunion, South Sudan, Djibouti, Seychelles, and Mauritius were omitted because of no history of DHS conduction. Additionally, Eritrea and Sudan were also not included due to the long period since their last conduction of DHS, i.e. Eritrea in 2002 and Sudan in 1989/90 (Table 1). It was conducted using the principle of a two-stage stratified sampling procedure. In the first stage, Enumeration Areas (EAs) were randomly selected proportionally to their respected clusters. In the second stage, households were selected. The primary objective of conducting DHS is to provide up-to-date information about health and health-related indicators for planning, policy formulation, monitoring, and evaluation of population and health programs in the respective countries.

Variables were extracted after a deep literature review and appended together to assess the pooled prevalence of diarrhea and associated factors in East Africa among under five children. In this study, the children's dataset (KR file) was used. Ultimately, a total of 129,651(weighted) children under the age of five were encompassed in this study.

**Table 1. Survey years of each country with respective weighted sample size.**

| Country's name | Survey year | Weighted sample size |
|---|---|---|
| Burundi | 2016 | 12,774 |
| Ethiopia | 2016 | 10,337 |
| Kenya | 2014 | 18,517 |
| Comoros | 2012 | 3,030 |
| Madagascar | 2008 | 11,769 |
| Malawi | 2015 | 16,336 |
| Mozambique | 2011 | 10,722 |
| Rwanda | 2019 | 7,616 |
| Tanzania | 2017 | 9,268 |
| Uganda | 2016 | 14,153 |
| Zambia | 2018 | 9,183 |
| Zimbabwe | 2015 | 5,944 |

## Study variables

**Dependent variable.**   The outcome variable was binary, children who had diarrhea at any time during the 2 weeks preceding the interview. The response variable diarrhea is recoded as follows: Those mothers/caregivers who responded yes to the question "had diarrhea in the last two weeks?" were coded as 1 and those who answered no were coded as 0 [21].

**Independent variables.**   We sub-portioned the independent variables into two groups; level-1 (individual-level factors) and level -2(community-level factors).

**Level-1 factors.**   Child's age, child's sex, number of under five years children, immunization status, duration of breast feeding in months, age of the mother/caregiver, education of the mother, mother's working status, mass media exposure of the mother, household wealth status, type of latrine, type of drinking water source and timing of breast feeding initiation after birth were considered for this study.

**Level-2 factors.**   The place of residence, community level of poverty, and community-level of educational status were variables assigned as community-level factors. The variable community level of poverty and community-level of educational status were generated by aggregating individual level factors at the cluster/community level.

**Operational definition.**   *Media exposure*. This variable is composite which consisted of watching television, listening to the radio, and reading magazines. Watching television (those who watch television less than once a week, at least once a week and every day are coded as = yes, otherwise = no), frequency of listening to the radio (listening less than once a week, at least once a week and every day are coded as = yes, otherwise = no) and frequency of reading Newspaper or magazine (reading less than once a week, at least once a week and every day are coded as = yes, otherwise = no) [22].

*Visits to health facility or visited by health worker*. Women either visited by health worker or had visited health facility in the last 12 months are categorized under "yes" and those who neither visited health facility nor visited by health worker were categorized under "no".

*Type of toilet*. Population using toilet characterized by flushing to somewhere else, pit latrine—without slab, bucket toilet, hanging toilet or other toilet were coded as "unimproved toilet" and population using toilet which flush—to piped sewer system, flush—to septic tank, flush—to pit latrine, flush—don't know where, pit latrine—ventilated improved pit, pit latrine —with slab or composting toilet were coded as "improved toilet" [22].

*Drinking water type.* Household using drinking water which is, piped into dwelling piped to yard/plot public tap/standpipe, piped to a neighbor, tube well or borehole, protected well, protected spring, rainwater, tanker truck, cart with small tank or bottled water were coded as "improved drinking water" and household categorized under unprotected well, unprotected spring, surface water or other sources of drinking water was coded as "unimproved drinking water" [22].

*Timing of BF initiation.* Children who initiated BF within one hour of birth are labeled as "early" and coded 1, apart from that labeled as "delayed" and coded as 0 [23].

*Community level of poverty.* Proportion of households assigned to poorest and poorer wealth index. Those fall at the median value and above are categorized under the high poverty level, and those who fall below the median value of the variables are categorized under the low poverty level. Median is used as a cut point because of skewed distribution. The same way of categorization was used for community-level educational status.

*Community-level of educational status.* Proportion of mother's/caregiver's of the child who is educated primary and above are categorized as having "high level of educational status" and otherwise "low level of educational status".

*Perceived distance from health facility.* The DHS program asks caregivers or mothers their perception whether the distance from health facility is a "big problem "or "no a big problem" when they were seeking medical advice or treatment for themselves when they are sick.

*Immunization status.* Fully vaccination definition is adopted from the number of children aged 12–23 months who received one dose of BCG vaccine, three doses of polio vaccine, three doses of pentavallent vaccine (DTP-hepB-Hib), three-dose of pneumococcal conjugate vaccine (PCV), two-doses of virus vaccine, and one dose of measles vaccine was considered as "fully vaccinated" otherwise "not fully vaccinated [24].

## Data analyses

Cross tabulations and summary statistics were done using STATA version 16 software. The forest plot technique was utilized to display the prevalence of diarrhea across countries. To plot 95% CI of the coefficient of each variable of the best-fitted model, STATA command "coefpot" was applied. AS the DHS datasets have hierarchical nature (sample is not taken randomly), non-independencies of observations and violation of equal variance assumption of the single level statistical model like logistic regression are inevitable.

In the multistage stratified clustered sampling of DHS, children within a cluster are more likely to relate to certain characteristics as compared to children between the clusters. To overcome those problems, to draw reliable inferences, we calibrated so what sophisticated model called the multilevel logistic model to identify factors associated with diarrhea. We first calibrated the null model (model with only constant/intercept) in order to declare nesting of observation within clusters and to determine the use of multilevel analysis. To warrant the use of multilevel analysis, ICC (intra-class coefficient) was checked. Intra-class coefficient takes the value between 0 and 1. If the intraclass coefficient value approaches value one, then it indicates observations within the cluster are more similar than observations between clusters. Therefore, it implies that a multilevel model is necessary for that specific dataset. It also shows how much of the response's total variation is explained by clustering.

Deviance Information Criterion (DIC), Log-Likelihood Ratio (LLR), Akaike information criteria (AIC), and Bayesian information criteria (BIC) were used as a model comparison and selection parameters. The model with the lowest values of those parameters was selected as the best-fitted model. The model comparison was done among the null model (a model with no independent variables), model I (a model with only individual-level factors), model II (a

model with only community-level factors) and model III (a model with both individual and community level independent variables). Variables with a p-value <0.2 in the bi-variable analysis were considered in the multivariable mixed-effect logistic regression model. Adjusted Odds Ratios (AOR) with a 95% Confidence Interval (CI) and p-value $\leq$ 0.05 in the multivariable model were used to declare significant factors associated with diarrhea.

## Ethical consideration

This study used datasets of national representative demographic health surveys. Therefore, ethical is approval not required. But, datasets for this study were requested by providing a clear explanation about the objectives and necessity of this study. We registered and requested the DHS dataset to the online database (www.dhsprogram.com) and received an authorization letter to download the requested datasets.

## Results

### Characteristics of the study population

The majority of the study participants were from Kenya (14.3%), Malawi (12.6%), Uganda (11%), and Burundi (9.9%%). The median age of the child was 28 months, with an interquartile range (IQR) of 13 to 43 months. Most of the children (78.4%) were from rural and more than half were born from primary level educated women (Table 2).

**Table 2. Characteristics of the study population in East Africa (N = 129,651).**

| Characteristics | Weighted frequency | Percent |
|---|---|---|
| **Country** | | |
| Burundi | 12,774 | 9.85 |
| Ethiopia | 10,337 | 7.97 |
| Kenya | 18,517 | 14.28 |
| Comoros | 3,030 | 2.34 |
| Madagascar | 11,769 | 9.08 |
| Malawi | 16,336 | 12.6 |
| Mozambique | 10,722 | 8.27 |
| Rwanda | 7,616 | 5.87 |
| Tanzania | 9,268 | 7.15 |
| Uganda | 14,153 | 10.92 |
| Zambia | 9,183 | 7.08 |
| Zimbabwe | 5,944 | 4.58 |
| **Age of child in months** | | |
| 0–6 | 11,315 | 12.5% |
| 7–12 | 9,895 | 11.0% |
| 13–24 | 18,690 | 20.7% |
| 25–59 | 50,363 | 55.8% |
| Sex **of the child** | | |
| Male | 45,441 | 50.3% |
| Female | 44,822 | 49.7% |
| **Immunization status** | | |
| **Fully immunized** | 52,436 | 58.5% |
| **Not fully immunized** | 37,206 | 41.5% |
| **Maternal age** | | |

*(Continued)*

**Table 2.** (Continued)

| Characteristics | Weighted frequency | Percent |
|---|---|---|
| 15–24 | 25,984 | 28.8% |
| 25–34 | 43,609 | 48.3% |
| 35–49 | 20,670 | 22.9% |
| **Maternal educational status** | | |
| None | 23,191 | 25.7% |
| Primary | 45,620 | 50.5% |
| Secondary &above | 21,452 | 23.8% |
| **Wealth status** | | |
| Poorest | 21,723 | 24.1% |
| Poorer | 19,555 | 21.7% |
| Middle | 17,428 | 19.3% |
| Richer | 16,712 | 18.5% |
| Richest | 14,845 | 16.5% |
| **Perceived distance from health facility** | | |
| Perceived as big problem | 36,019 | 44.5% |
| Perceived as Not big problem | 44,864 | 55.5% |
| **Types of latrine** | | |
| Unimproved toilet | 26578 | 40.4% |
| Improved toilet | 39288 | 59.6% |
| **Source of drinking water** | | |
| Unimproved | 31,305 | 35.4% |
| Improved | 57,405 | 63.6% |
| **Being Twin** | | |
| No | 87,937 | 97.3% |
| Yes | 2,473 | 2.7% |
| **Mother's Current working status** | | |
| No | 33,103 | 40.9% |
| Yes | 47,803 | 59.1% |
| **No. of under five years children** | | |
| ≤2 | 73,621 | 81.6% |
| >2 | 16,641 | 18.4% |
| **Time of breast feeding initiation after birth** | | |
| Immediately to 1 hour | 55,910 | 62.0% |
| 1 hour | 34,352 | 38.0% |
| **Media exposure** | | |
| Exposed | 59,387 | 65.8% |
| Not exposed | 30,876 | 34.2% |
| **Community level factors** | | |
| **Place of residency** | | |
| Urban | 21,507 | 23.8% |
| Rural | 68,756 | 76.2% |
| **community level educational status** | | |
| Low level | 43998 | 48.7% |
| high level | 46264 | 51.3% |
| **Community level of poverty** | | |
| Low level | 46435 | 51.4% |
| High level | 43828 | 48.6% |

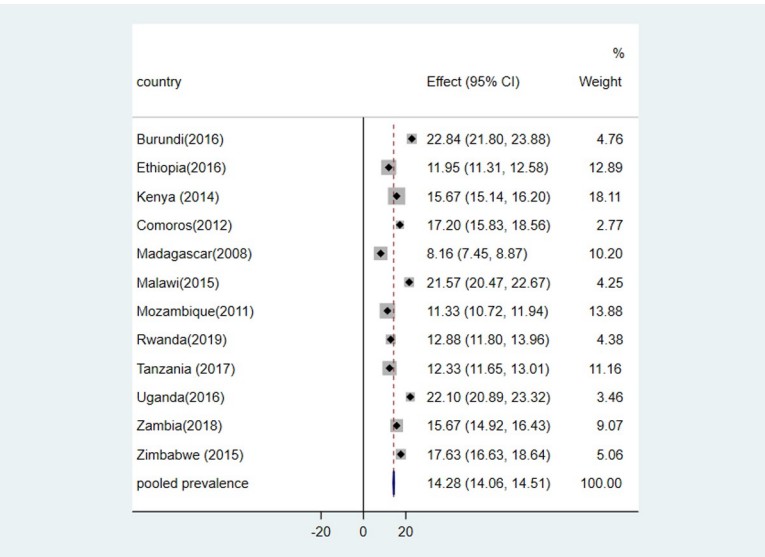

**Fig 1. Forest plot of the pooled prevalence of diarrhea among under five years of children in Eastern African countries.**

## Prevalence of diarrhea in East Africa

The pooled prevalence of diarrhea among under five years of children in East Africa was 15.86% [95% CI: 15.66, 16.06]. The highest prevalence of diarrhea among under five years children was observed in Burundi with a prevalence rate of 22.84% [95%CI: 22.80%, 23.88%] followed by Uganda (22.10%) and Malawi (21.57%). Conversely, the lowest prevalence was also noticed in Madagascar (8.16%), Ethiopia (11.33%), and Tanzania (12.33%). The estimated prevalence of diarrhea is significant in all 12 countries because of the estimated prevalence of each country didn't overlap with pooled regional prevalence estimate (Fig 1).

The weighted percentage on the right side of the forest plot indicates the influence of each country's studies on the pooled prevalence of diarrhea. Accordingly, Kenya, Mozambique, Ethiopia, and Tanzania were the most influential countries on the pooled prevalence of diarrhea in children. The broken vertical line of the forest plot stands for the pooled prevalence of diarrhea with the confidence interval corresponding to the width of the diamond. The diamond to the left and right of this broken line indicates the point prevalence of diarrhea in the corresponding country is lower and higher than the pooled prevalence of diarrhea, respectively. A point crossing the broken line denotes that overlapping of confidence interval with pooled prevalence. The unbroken vertical line in this study indicates no prevalence of diarrhea (zero prevalence).

## Sensitivity analysis

To declare the stability of those compiled studies, sensitivity analysis was conducted (Table 3). In the sensitivity analysis, each analysis is produced by excluding a single study (country in this case). This analysis is performed to view outlier studies (studies that yield exaggerated effect size) while studying by pooling more than one study. Including such outliers, studies distort overall results. From Table 2 we can illustrate that Ethiopia's point estimate of the prevalence of diarrhea is outside the combined confidence interval. This means, omitting Ethiopia from the study will increase the pooled prevalence of diarrhea to 25%. Hence, the rest studies'

**Table 3. Sensitivity analysis of the pooled prevalence of diarrheal diseases among pregnant women in East Africa.**

| Study omitted | point Estimate | 95% CI | |
|---|---|---|---|
| | | LCIL | UCIL |
| Burundi(2016) | 12.99 | 2.99 | 22.99 |
| Ethiopia(2016) | 24.92 | 8.46 | 41.38 |
| Kenya (2014) | 13.65 | 2.67 | 24.62 |
| Comoros(2012) | 12.92 | 2.89 | 22.95 |
| Madagascar(2008) | 12.89 | 2.80 | 22.98 |
| Malawi(2015) | 13.03 | 3.05 | 23.01 |
| Mozambique(2011) | 12.89 | 2.80 | 22.98 |
| Rwanda(2019) | 12.94 | 2.92 | 22.96 |
| Tanzania (2017) | 13.00 | 3.02 | 22.99 |
| Uganda(2016) | 12.99 | 2.99 | 22.98 |
| Zambia(2018) | 13.10 | 3.14 | 23.05 |
| Zimbabwe (2015) | 12.90 | 2.84 | 22.96 |
| Combined effect | 13.41 | 3.48 | 23.34 |

point estimates are inside the confidence interval of the combined effect, we can conclude that those studies are stable.

## Model comparison

Measures of random effects/measure of variation included intraclass/cluster correlation (ICC), median odds ratio (MOR), and proportional change in variance (PCV). ICC was calculated to ensure intra-cluster variability of the study participants. Children from the same cluster are highly likely to share common characteristics than children outside the cluster. Output from the intercept-only model (null model) uncovered that the intraclass correlation coefficient is 1.6%. This indicates that 1.6% of the variation of diarrheal diseases is contributed by the difference between clusters.

The median odds ratio generated from the null model also shows the inter-cluster variation of diarrheal diseases. The median odds ratio value (1.25) generated by the null model is interpreted as; when two individuals having the same characteristics (covariates) picked from different clusters randomly, the individual from higher risk cluster had 24% higher odds of encountering diarrheal diseases compared to the individual from the low risk cluster. Additionally, the proportional change in variance from model III (full model) illustrates that28.3% of the odds of childhood diarrhea across the selected East African countries was accounted by both individual and community-level factors (Table 4).

According to Table 4, model III (model embodies both individual-level factors and community-level factors) is best the fitted model for this dataset. This is because model three (full model) is with the lowest BIC and deviance.

## Factors associated with diarrheal diseases

From the bivariable multilevel modeling, variables except for toilet type and media exposure all variable showed significant association with the dependent variable at p-value <0.2 (Fig 2).

The odds of diarrhea were 41% [AOR = 1.41, 95% CI: 1.33, 1.49] and 17% [AOR = 1.1.17, 95% CI: 1.10, 1.23] higher in children whose mother's age is 15–24 and 25–34 years respectively as compared to 35–49 years of age mother. Similarly, the odds of diarrhea was 3.10 [AOR = 3.10, 95% CI: 2.86, 3.35] and 2.56 times [AOR = 2.56, 95% CI: 2.38, 2.75] higher in

**Table 4. Model comparison and fitness parameter outputs.**

| Fitness parameter | Null model | model I | model II | model III |
|---|---|---|---|---|
| Community level variance | .053 | 0.039 | 0.051 | 0.038 |
| ICC | 1.6% | 1.17% | 1.54% | 1.16% |
| MOR | 1.25[1.21,1.29] | 1.20[1.17,1.25] | 1.241[1.20–1.281] | 1.20[1.17,1.25] |
| PCV (%) | Reference | 26.4% | 3.8%% | 28.3% |
| **Model fitness** | | | | |
| Log- likelihood ratio(LLR) | -55586 | | | |
| DIC(-2LLR) | 111172 | | | |
| AIC | 111176.1 | | | |
| BIC | 111195.7 | | | |

**NOTE:** ICC, intra-cluster correlation; MOR, median odds ratio; DIC, deviation information criterion. Null Model is the empty model, baseline model without any determinant variable. Model I is adjusted for individual-level factors. Model II is adjusted for community-level factors. Model III is the final model adjusted for both individuals and community-level factors.

children whose age is 7–12 and 12–24 months respectively as contrasted to children aged 0–6 months. Conversely, the odds of diarrhea is 12% lower in children aged 25–59 months compared to children aged 0–6 months. Regarding the sex of the child, being male raises the odds of diarrhea by 8% [AOR = 1.08, 95%CI: 1.03, 1.12] higher than compared to counterpart (Table 5).

Another pertinent finding was the timing of breastfeeding initiation after birth. The odds of diarrhea is 1.15 [AOR = 1.15, 95% CI: 1.10, 1.20] times higher in children who initiated breastfeeding after one hour of birth compared to those who initiated within one hour of birth. Additionally, for children belonging to poor and middle wealth households, the odds of diarrhea increase by 16% [AOR = 1.16, 95% CI: 1.09, 1.23] and 13% [AOR = 1.13, 95% CI: 1.06, 1.21] respectively in contrast to children from rich household.

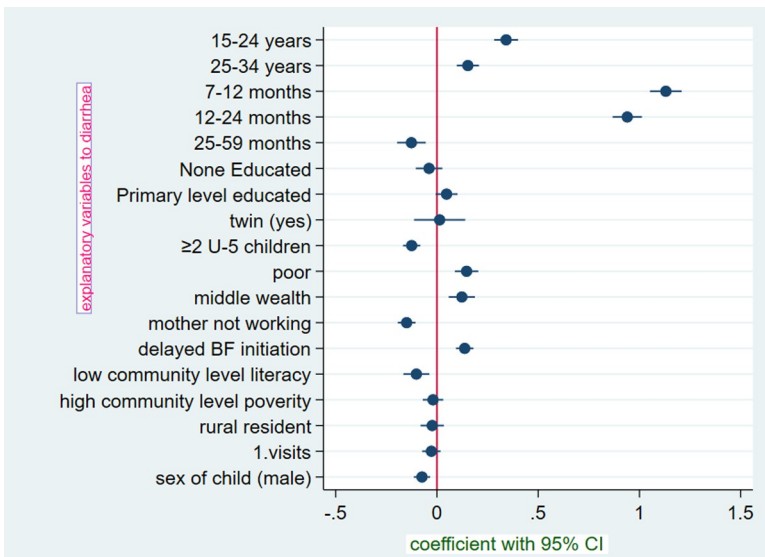

**Fig 2. Coefficient plot of both community and individual-level factors of diarrhea among under five years children in East Africa with respective 95% CI.**

**Table 5. Multivariable multilevel logistic analysis result of diarrhea diseases among under five years children in East African countries.**

| Characteristics | Model I (95%CI AOR) | Model II (95%CI AOR) | Model III ((95%CI AOR) |
|---|---|---|---|
| **Maternal age** | | | |
| 35–49 years | Ⓡ | | Ⓡ |
| 15–24 years | **1.41[1.33,1.49]*** | | **1.41[1.33, 1.49]*** |
| 25–34 years | **1.17[1.10,1.23]*** | | **1.17[1.10, 1.23]*** |
| **Child's age** | | | |
| 0–6 months | Ⓡ | | Ⓡ |
| 7–12 months | 3.10[2.87,3.35]* | | **3.10[2.86, 3.35]*** |
| 12–24 months | 2.56[2.38,2.75]* | | **2.56[2.38, 2.75]*** |
| 25–59 months | 0.88]0.82,0.95]* | | **0.88[0.82, 0.95]*** |
| **Maternal Educational status** | | | |
| Secondary &above | Ⓡ | | Ⓡ |
| None | 0.94[0.89,1.01] | | 0.96[0.90, 1.03] |
| Primary | 1.04[0.99,1.10] | | 1.05[0.99, 1.11] |
| **Twin** | | | |
| No | Ⓡ | | Ⓡ |
| Yes | 1.01[0.89, 1.15] | | 1.01[0.89, 1.15] |
| **No. of under five years children** | | | |
| < 2 child | Ⓡ | | Ⓡ |
| ≥2 children | **0.88[0.84, 0.92]*** | | **0.88[0.85, 0.92]*** |
| **Wealth index** | | | |
| Rich | Ⓡ | | Ⓡ |
| Poor | **1.14[1.09, 1.21]*** | | **1.16[1.09, 1.23]*** |
| Middle | **1.12[1.06, 1.19]*** | | **1.13[1.06, 1,21]*** |
| **Mother Working status** | | | |
| Working | Ⓡ | | Ⓡ |
| Not working | **0.86[0.82, 0.90]*** | | **0.86[0.82, 0.90]*** |
| **Timing breast feeding initiation** | | | |
| Within one hour | Ⓡ | | Ⓡ |
| Delayed initiation | **1.15[1.10, 1.20]*** | | **1.15[1.10, 1.20]*** |
| **Visit to health facility or visited by health worker** | | | |
| Yes | Ⓡ | | Ⓡ |
| No | 0.97[0.93, 1.01] | | 0.97[0.93, 1.02] |
| Sex of child | | | |
| Female | Ⓡ | | Ⓡ |
| Male | **1.08[1.03, 1.12]*** | | **1.08[1.03, 1.12]*** |
| **Community level factors** | | | |
| **Residency** | | | |
| Urban | | Ⓡ | Ⓡ |
| Rural | | 1.02[0.98, 1.07] | 0.98[0.92, 1.03] |
| **Community level educational status** | | | |
| High | | Ⓡ | Ⓡ |
| Low | | **1.15 [0.09, 1.22]** | **1.10[1.03, 1.18]** * |
| **Community level poverty** | | | |
| Low | | Ⓡ | Ⓡ |
| High | | 1.02 [0.97, 1.07] | 0.98[0.93, 1.03] |
| Constant | 0.098[0.09, 0.11] | 0.19 [0.18, 0.20] | 0.11[0.10, 0.12] |

**NOTE:** Ⓡ- reference, AOR- adjusted odds ratio.

The odds of diarrheal diseases decrease by 12% [AOR = 0.88 95%CI: 0.85, 0.92] and 14% [AOR = 0.86 95%CI: 0.82, 0.90] in children from households having two and more children and in children whose mother were not working respectively. The odds of diarrhea is 1.1 times higher in children who was born from a community of low level of education relative to a high level.

## Discussion

The prevalence of diarrhea among under-five years of age children is high in East Africa. This high prevalence is may due to the low level of sanitation in East African countries [13]. The conjoint (pooled) prevalence of diarrhea in the nominated East African countries is 14.28% [14.06%, 14.51%]. The pooled prevalence of diarrhea diseases among children generated from this study is consistent with the study done in India [25] and lower than the study in Egypt (19.5%) [26], Ghana (19.2%) [27], India (25.2%) [28] and cross-sectional survey from south Wollo district, Ethiopia (23.1%) [29]. However, this finding is higher than research done in Nigeria (12.7%) [30]. This dissimilarity is possibly attributable to a difference in socio-demographic characteristics, location, climate, culture of stool disposal, water access, and a culture of handwashing.

Being young-aged cohort of mothers, compared to an old aged cohort of mothers (i.e. 35–49 ages), was found to be elevated the probability of prevalence of diarrhea among under five years children. This finding is in agreement with the study done in Uganda [7] and Nepal [31]. This might be explained by; older mothers having good knowledge and experiences about child health in general and diarrheal diseases specifically [32]. Government effort to educate young mothers is therefore called for.

The odds of developing diarrheal diseases is higher in children who were aged 7–12 and 13–24 months compared to children aged 0–6 months. This finding is in agreement with other studies [7, 14–16, 33]. This high odds of childhood diarrhea prevalence might attribute to the start of supplementary feeding after the age of six months. Children who started supplementary feeding have a high probability of feeding unhygienic foods that might have paved the way to diarrheal diseases. After the age of 6 months children, due to the development of hand-mouth coordination, are highly likely to bring the infectious agent to their mouth and are expected to increase the episode of diarrhea diseases at this age. Conversely, the odds of diarrheal disease among children aged 25–59 months is reduced by 12% as compared to children aged 0–6 months. The possible explanation for this result is, old age children are more potential to hold out against diarrheal diseases because of the more developed immunity compared to younger children.

In this study, it was found that families who had two under-five years children or above were less likely to have diarrhea than those who had only one child. This finding is contrary to previous studies which had suggested that, as the number of under five children increases, the probability of occurrence of diarrheal diseases is high [34, 35]. This inconsistency may be due to those households with a high number of under-five children being more familiar with the sequel of childhood diarrheal diseases; the more they experienced and knowledgeable in knowing ways of transmission of diarrheal diseases, the less likely to have diarrhea in the subsequent child.

Childhood diarrheal disease was statistically associated with household wealth status. Children from poor and middle-income households are at high odds of diarrheal diseases compared to children from rich households. Many studies also fortify this evidence [7, 15, 17, 18]. This high odds of diarrheal diseases in those groups of children can be explained by, economically marginalized families being doubtful to bring their sick child to health facilities due to

concern of transport fees and cost of health services. Another possible justification for this finding is that children from the poor household are less likely to get a balanced diet and more likely to encounter malnutrition that precipitates and elongate the duration of diarrhea [36, 37]. The policy implication of this finding is, government and other concerned bodies might have to intervene malnutrition to cease diarrheal diseases.

Delayed breastfeeding initiation was positively associated with diarrhea (increases the likelihood of diarrhea). Children who begin their first breastfeeding after one hour of birth have higher odds of diarrheal episode compared to their counterparts. The finding is in accord with a recent study [7], indicating that children breastfed in the first half an hour after birth reduce the probability of diarrhea. Additionally, being children born from mothers belonging to low community-level educational status and being male are at high probability of experiencing diarrhea as compared to counterparts. A similar declaration was also made by other studies [8, 16, 18, 38, 39]. Lastly, children of not working mothers have a 12% lower odds to have diarrhea than those of mothers who work. This might be elucidated by, mother who always present at household had more time to take care of the health of her child.

## Strengths and limitations of the study

Finally, several strengths and limitations can be drawn from this study. The main strength of this study is, as we compiled several datasets from different countries which used almost the same data collection tools, the statistical power of this study is reliable, and the generalizability of this study from such a huge sample size is trustworthy. In addition to this, the hierarchical nature of the surveys was considered through conducting an advanced statistical model (multilevel). The main limitation of this study is that due to the cross-sectional study of DHS, causation can't be assured. Social desirability bias is inevitable. Additionally, the variable types of water sources were dropped from this study because of the high number of missing values and this might have over or underestimated our model performance. Hence, we used data from a secondary survey, other pertinent behavioral and cultural factors are not embodied in this study.

## Conclusions

The pooled prevalence of diarrhea among under five years of children in the 12 East African countries is high. This may raise some difficulty in achieving the Sustainable development goal (SDG-3) that targeted reducing under-five mortality to less than 25 deaths per 1000 live births as diarrhea is the main cause of under-five death [40]. Maternal age, child's age, wealth status of households, the timing of breastfeeding initiation, sex of a child, community level of educational status, working status of mothers, and the number of under-five children were factors that were associated with diarrheal diseases among under-five children in East African countries. Scaling up of maternal and child health services by government and other concerned bodies should consider those economically marginalized communities. Additionally, awareness should be created for those uneducated mothers concerning the nature of childhood diarrhea.

## Acknowledgments

The authors acknowledge MEASURE DHS for permitting us to access and download the East African DHS datasets.

## Author Contributions

**Conceptualization:** Abiyu Abadi Tareke, Ermias Bekele Enyew.

**Data curation:** Abiyu Abadi Tareke, Ermias Bekele Enyew.

**Formal analysis:** Abiyu Abadi Tareke, Bayley Adane Takele.

**Funding acquisition:** Abiyu Abadi Tareke, Bayley Adane Takele.

**Investigation:** Abiyu Abadi Tareke.

**Methodology:** Abiyu Abadi Tareke.

**Project administration:** Abiyu Abadi Tareke.

**Resources:** Abiyu Abadi Tareke, Ermias Bekele Enyew, Bayley Adane Takele.

**Software:** Abiyu Abadi Tareke.

**Supervision:** Abiyu Abadi Tareke, Ermias Bekele Enyew, Bayley Adane Takele.

**Validation:** Abiyu Abadi Tareke, Bayley Adane Takele.

**Visualization:** Abiyu Abadi Tareke, Ermias Bekele Enyew, Bayley Adane Takele.

**Writing – original draft:** Abiyu Abadi Tareke, Bayley Adane Takele.

**Writing – review & editing:** Abiyu Abadi Tareke, Ermias Bekele Enyew.

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
