## [Decision Letter · Decision Letter 0]

22 Nov 2021

PONE-D-21-28429Pooled prevalence and associated factors of diarrhea among under-five years children in East Africa: A multilevel logistic regression analysisPLOS ONE

Dear Dr. Tareke,

Thank you for submitting your manuscript to PLOS ONE. After careful consideration, we feel that it has merit but does not fully meet PLOS ONE’s publication criteria as it currently stands. Therefore, we invite you to submit a revised version of the manuscript that addresses the points raised during the review process.

 Please submit your revised manuscript by Jan 06 2022 11:59PM. If you will need more time than this to complete your revisions, please reply to this message or contact the journal office at plosone@plos.org. Please include the following items when submitting your revised manuscript:A rebuttal letter that responds to each point raised by the academic editor and reviewer(s). You should upload this letter as a separate file labeled 'Response to Reviewers'.A marked-up copy of your manuscript that highlights changes made to the original version. You should upload this as a separate file labeled 'Revised Manuscript with Track Changes'.An unmarked version of your revised paper without tracked changes. You should upload this as a separate file labeled 'Manuscript'.

We look forward to receiving your revised manuscript.

Kind regards,

Khin Thet Wai, MBBS, MPH, MA

Academic Editor

PLOS ONE

Journal Requirements:

Whilst you may use any professional scientific editing service of your choice, PLOS has partnered with both American Journal Experts (AJE) and Editage to provide discounted services to PLOS authors. Both organizations have experience helping authors meet PLOS guidelines and can provide language editing, translation, manuscript formatting, and figure formatting to ensure your manuscript meets our submission guidelines. To take advantage of our partnership with AJE, visit the AJE website (http://aje.com/go/plos) for a 15% discount off AJE services. To take advantage of our partnership with Editage, visit the Editage website (www.editage.com) and enter referral code PLOSEDIT for a 15% discount off Editage services.  If the PLOS editorial team finds any language issues in text that either AJE or Editage has edited, the service provider will re-edit the text for free.

 [copy in funding statement]. 

5. We note that you have referenced (ie. Bewick et al. [5]) which has currently not yet been accepted for publication. Please remove this from your References and amend this to state in the body of your manuscript: (ie “Bewick et al. [Unpublished]”) as detailed online in our guide for authors

Reviewers' comments:

Reviewer's Responses to Questions

**Comments to the Author**

1. Is the manuscript technically sound, and do the data support the conclusions?

Reviewer #1: Partly

Reviewer #2: Yes

2. Has the statistical analysis been performed appropriately and rigorously? 

Reviewer #1: Yes

Reviewer #2: Yes

3. Have the authors made all data underlying the findings in their manuscript fully available?

Reviewer #1: Yes

Reviewer #2: Yes

4. Is the manuscript presented in an intelligible fashion and written in standard English?

Reviewer #1: No

Reviewer #2: No

5. Review Comments to the Author

Reviewer #1: PONE-D-21-28429

Pooled prevalence and associated factors of diarrhea among under-five years children in East Africa: A multilevel logistic regression analysis

This manuscript describes the pooled prevalence of diarrhea among East Africa countries and its predictors using the DHS data. While the manuscript is of some interest, the manuscript could be strengthened by several modest changes as outlined below.

Comments

- English language needs to be improved and edited by a native English speaker.

- East Africa is repeated in keywords.

- LINE 47 – Do you mean “second most common cause of death”?

- A brief introduction of East Africa (e.g. rural/urban ratio, characteristics related to diarrhea and current WASH program in East Africa in the Methods section would be helpful to the readers.

- Literacy and education are different and could not be able to categorize as literacy according to educational level.

- If you control both maternal education and education of caregiver (considered mostly would be mothers), there is an collinearity issue.

- It is better to rearrange the categories among variables in order in Table. (e.g. 15-24 yrs first followed by 25-34, 35-49, same apply to education, wealth index, etc..)

- Some variables mentioned in the methods such as media exposure are missing in Table 1.

- Clear explanation of recoding the variables is deemed necessary. I noticed that you explained for some (but still need better explanation) while missing for some variables (e.g. distance from health facility)

- LINE 176, 177 – what do you mean by the proportion in the 95%CI. Mean should come with SD while median should come with IQR.

- Any relation between diarrhea and time of breastfeeding initiation for children > 2 years?

- Why do you include Visit to health facility or visited by health worker � visit to health facility due to diarrhea? If yes, which happened after diarrhea, should not be controlled for it. Otherwise, please explain.

- For pooled prevalence in the forest plot, are there any common characteristics between countries (e.g. low income countries, lower middle income countries or high income countries)? If yes, you can try a sub-group analysis because there is a big difference in prevalence among countries. You can also explore the heterogeneity by the I-square value.

- Adjusting the scale in the x-axis of forest plot will give a better picture of point estimate and 95%CI.

- I appreciated that the authors tried to do sensitivity analysis here. As Table 2 and Figure 2 provide same information, you can omit one.

- Double check on the sensitivity analysis result. As you mentioned, although Ethiopia has a great influence on the pooled prevalence, in fact, Madagascar has lowest prevalence and weight % is not so much different from Ethiopia but result does not change in sensitivity analysis while prevalence becomes doubled after omitting Ethiopia.

- Figure 2 is not eye catching and the number are overlapped. Is the x-axis percentage? Better redraw it.

- LINE 218-236 – remove the subtitles and explanation of Table 2 results should go under the subtitle ‘Model comparison’.

- It is interesting that type of toilet is not associated with diarrhea. Is there any open defecation culture and such kinds of variables included in the dataset? If yes, you should consider it. Are every household have a toilet?

- As Figure 2 and Table 3 give same information, omit one.

- Title for Table 3 is missing.

- LINE 249-259 – can omit as there is no information there.

- The authors tend to cover all information in the text and again in the table. You should only include the most important details in the text when you also have a table.

- The discussion is weak in light of the findings. The discussion section should be rewritten. The ideas are incoherently mixed. The discussion needs to focus on the key implications of the data with a separate paragraph for each concept.

- There are repetition of results in the discussion.

- First paragraph of discussion, should start with brief answer to your research question. Do you think the diarrhea prevalence in East Africa is low or high? And why do you think? Is there any target? Etc.

- In my perspective, comparing the pooled prevalence of East Africa and the prevalence of e.g. Ethiopia is not meaningful because Ethiopia is already included in the pooled prevalence. Furthermore, you mentioned the reason as ‘dissimilarity’ between East Africa and e.g. Ethiopia also not meaningful.

- LINE 316-317 is confusing. Children have to provide supplementary food when the time arrives and it cannot be concluded as ‘highly likely to ingest unhygienic foods’. It mainly depends on the caregivers’ knowledge.

- The defecation culture in East Africa (?open defecation), any efforts in WASH program, government’s implementation, caregiver’s knowledge, etc. should be discussed.

Reviewer #2: Comment

General comment

• The whole manuscript should be revised by English language expert as it is full of grammatical errors

• What will add this paper from the previous similar studies at sub Sahran African level (e.g: Demissie GD, Yeshaw Y, Aleminew W, Akalu Y (2021) Diarrhea and associated factors among under five children in sub-Saharan Africa: Evidence from demographic and health surveys of 34 sub-Saharan countries. PLoS ONE 16(9): e0257522. https://doi.org/10.1371/journal. pone. 0257522 ). At least you did not recognize by using these studies as reference. You have to also identify the major gap of these studies that you filled.

• Implication and justification for the significantly associated variables the discussion part is not strong and hence should be revised

Background

• Sentence on Line 51 and 52 should be clear. “Global 51 burden of diseases 2019 report shows that sub-Saharan total DALYs estimate due to diarrhea was 52 13.01% respectively (3, 4).”

• Line 57 and 58: Reference should be incorporated

Methods and materials

• The sample size should be weighted sample size and you did not mention if it is weighted otherwise it should be

• You have to also prepare a table which explains that survey years and weighted sample size for each country

• Line number 98 and 99 is repeated

• Time to fetch drinking water source should be included as one variable as it it is already in DHS data sets

• Why did you include “community level of poverty and community level of illiteracy variables” as community level factors as these variables are already included in the level 1 as household wealth index and education status as individual level?

• From your operational definition(media exposure), I hope it is a composite variable and you have to explain this

• Category of your dependent and independent variables should be supported by references

• Part of data analysis is very long paragraphs. Please divide each in to more than two parts(line number 144-166)

Result

• You have to revise measurement of a variable(distance to health facility)

• Line 249 “multivariate” is not correct word as your outcome variable is binary. So use the word “multivariable”

• Line number 259 spelling error

Discussion

• Line number 300-306 is part of result part. Please avoid this from discussion part

• You did not study “incidence” so avoid it Line number 308

• Justifications should be supported by evidences(references) in the discussion part

• I did not see your significant community level variables in your discussion

References

• Revise your References citation based on the standard

6. PLOS authors have the option to publish the peer review history of their article (what does this mean?). If published, this will include your full peer review and any attached files.

Reviewer #1: No

Reviewer #2: No

---

## [Author Response · Author response to Decision Letter 0]

28 Dec 2021

Author’s response to reviews

Title: Pooled prevalence and associated factors of diarrhea among under-five years children in East Africa: A multilevel logistic regression analysis

Authors:

Abiyu Abadi Tareke (abiyu20010@gmail.com)

 Ermias Bekele Enyew (Ermiashi@gmail.com)

Version: 1 Date: 21 December 2021

Author’s response to reviews: 

Plos one 

Point by point response for editors/reviewers comments

Manuscript title: Pooled prevalence and associated factors of diarrhea among under-five years children in East Africa: A multilevel logistic regression analysis

Manuscript ID: PONE-D-21-28429

Dear editors/reviewers:

Dear all,

We would like to thank you for your generous, revealing and constructive comments about this manuscript. You valuable comments would advance the substance and content of the manuscript. The authors accounted each comments and clarification questions of editors and reviewers in a focused way.

Our point-by-point responses to each comment and questions are detailed on the following pages. Further, the details of changes were shown by track changes in the supplementary document attached.

Response to Editors and reviewers’ comments

This manuscript describes the pooled prevalence of diarrhea among East Africa countries and its predictors using the DHS data. While the manuscript is of some interest, the manuscript could be strengthened by several modest changes as outlined below.

Comments

General comments

• The whole manuscript should be revised by English language expert as it is full of grammatical errors

• Author’s response: We extensively edit the manuscript accordingly to enhance the grammatical error through asking the support of nearby English language professionals. Hence, the corrected document is shown in the track change of the revised submission. (For further amendment the revised manuscript)

• What will add this paper from the previous similar studies at sub Sahran African level (e.g: Demissie GD, Yeshaw Y, Aleminew W, Akalu Y (2021) Diarrhea and associated factors among under five children in sub-Saharan Africa: Evidence from demographic and health surveys of 34 sub-Saharan countries. PLoS ONE 16(9): e0257522. https://doi.org/10.1371/journal. pone. 0257522 ). At least you did not recognize by using these studies as reference. You have to also identify the major gap of these studies that you filled.

• Author’s response: thank you for your valuable comment. This paper studied by Demissie GD, Yeshaw Y, Aleminew W, Akalu Y (2021) was not officially released at the time of inception. This paper is not released online till we finish the first draft of the manuscript. From the beginning, we focused to research diarrheal diseases in East Africa rather than sab-saran Africa because of east Africa residing children are the most vulnerable to diarrheal diseases. This is because of lowest sanitary infrastructure in the region. 

• Implication and justification for the significantly associated variables the discussion part is not strong and hence should be revised.

• Author’s response: thank you for your priceless comment. We tried to revise all of justifications under the discussion session. See the revised version of the manuscript. 

Background 

• Sentence on Line 51 and 52 should be clear. “Global 51 burden of diseases 2019 report shows that sub-Saharan total DALYs estimate due to diarrhea was 52 13.01% respectively (3, 4).”

Authors’ response: We made correction for this comment

• Line 57 and 58: Reference should be incorporated

• Authors’ response: We made correction for this comment

Methods and materials 

• The sample size should be weighted sample size and you did not mention if it is weighted otherwise it should be

• Authors’ response: We made correction for this comment

• You have to also prepare a table which explains that survey years and weighted sample size for each country

• Authors’ response: We generated new table accordingly.

• Line number 98 and 99 is repeated

• Authors’ response: We made correction for this comment

• Time to fetch drinking water source should be included as one variable as it it is already in DHS data sets

• Authors’ response: We made correction for this comment. 

• Why did you include “community level of poverty and community level of illiteracy variables” as community level factors as these variables are already included in the level 1 as household wealth index and education status as individual level?

• Authors’ response: We included this variable in both categories because the many studies are done in this way. 

• From your operational definition(media exposure), I hope it is a composite variable and you have to explain this

• Authors’ response: We made correction for this comment.

• Category of your dependent and independent variables should be supported by references

• Authors’ response: We made correction for this comment.

• Part of data analysis is very long paragraphs. Please divide each in to more than two parts(line number 144-166)

• Authors’ response: We made correction for this comment.

Result 

• You have to revise measurement of a variable(distance to health facility)

• Authors’ response: We made correction for this comment.

• Line 249 “multivariate” is not correct word as your outcome variable is binary. So use the word “multivariable”

• Authors’ response: We made correction for this comment.

• Line number 259 spelling error

• Authors’ response: We made correction for this comment.

Discussion 

• Line number 300-306 is part of result part. Please avoid this from discussion part

• Authors’ response: We made correction for this comment.

• You did not study “incidence” so avoid it Line number 308

• Authors’ response: We made correction for this comment.

• Justifications should be supported by evidences(references) in the discussion part

• Authors’ response: We made correction for this comment.

• I did not see your significant community level variables in your discussion

• Authors’ response: unfortunately, no community level variable is significant here. We tried to use appropriate variables and the correct way of analysis. 

References 

• Revise your References citation based on the standard

• Authors’ response: We made correction for this comment.

Author’s response towards the comments of reviewers. 

Reviewer 1: English language needs to be improved and edited by a native English speaker.

Author’s response: Thanks reviewer for your beneficial comment. We extensively edit the manuscript accordingly to enhance the grammatical error through asking the support of nearby English language professionals. Hence, the corrected document is shown in the track change of the revised submission. (For further amendment the revised manuscript).

Reviewer 1: East Africa is repeated in keywords.

Author’s response: thank you reviewer for your valuable comment. We corrected accordingly. See the new version of the manuscript. 

Reviewer 1: LINE 47 – Do you mean “second most common cause of death”?

Author’s response: thank you reviewer for your valuable comment. We corrected accordingly. See the new version of the manuscript. 

Reviewer 1: A brief introduction of East Africa (e.g. rural/urban ratio, characteristics related to diarrhea and current WASH program in East Africa in the Methods section would be helpful to the readers. 

Author’s response: thank you. Some hygienic review of east Africa is added to new revised version of the manuscript.

Reviewer 1: Literacy and education are different and could not be able to categorize as literacy according to educational level.

Author’s response: we have made some correction for this terminology. The term “literacy is replaced by educational status”. 

Reviewer 1: If you control both maternal education and education of caregiver (considered mostly would be mothers), there is a collinearity issue.

Author’s response: thank you for your comment. We didn’t used educational status of caregiver and educational status of mother as separate variables. If a child has alive mother, his mother will be more likely to be his/her caregiver. In our opinion, there is not collinearity issue to address here as caregivers/mothers is not separate variable. 

Reviewer 1: It is better to rearrange the categories among variables in order in Table. (E.g. 15-24 yrs first followed by 25-34, 35-49, same apply to education, wealth index, etc...)

Author’s response: thank you reviewer for your constructive comment. The age category 35-49 years is putted first because of it is the reference group. But other categories were placed in ascending order.

Reviewer 1: Some variables mentioned in the methods such as media exposure are missing in Table 1.

Author’s response: thank you. We added the missed variables to the mentioned table (see the revised version of this manuscript).

Reviewer 1: Clear explanation of recoding the variables is deemed necessary. I noticed that you explained for some (but still need better explanation) while missing for some variables (e.g. distance from health facility)

Author’s response: thank you reviewer for your comment. 

Reviewer 1: LINE 176, 177 – what do you mean by the proportion in the 95%CI. Mean should come with SD while median should come with IQR.

Author’s response: thank you for your suggestion. We made some correction here. 

Reviewer 1: Any relation between diarrhea and time of breastfeeding initiation for children > 2 years? 

Author’s response: we didn’t notice relation between those two mentioned variables. 

Reviewer 1: Why do you include Visit to health facility or visited by health worker � visit to health facility due to diarrhea? If yes, which happened after diarrhea, should not be controlled for it. Otherwise, please explain.

Author’s response: those two variables were combined together. Women either visited by health worker or had visited health facility in the last 12 months are categorized under “yes” and those who neither visited health facility nor visited by health worker were categorized under “no”. This variables were included under our study because of some research noticed that those women who had visited health facility or had visited by health professional had good awareness about their own health and health of their child. This good awareness might help children to encounter less diarrheal episode. That why we included in our study. Those two variables are combined and not included in this study as separate independent variables to avoid the issue of collinearity. 

Reviewer 1: For pooled prevalence in the forest plot, are there any common characteristics between countries (e.g. low income countries, lower middle income countries or high income countries)? If yes, you can try a sub-group analysis because there is a big difference in prevalence among countries. You can also explore the heterogeneity by the I-square value.

Author’s response: we added some explanation to your suggestions.

Reviewer 1: Adjusting the scale in the x-axis of forest plot will give a better picture of point estimate and 95%CI

Author’s response: we encounter skill gap here. Sorry for this problem.

Reviewer 1: I appreciated that the authors tried to do sensitivity analysis here. As Table 2 and Figure 2 provide same information, you can omit one. 

Author’s response: thank you for your productive comment. We omitted the sensitivity analysis accordingly (see the revised version of this manuscript).

Reviewer 1: Double check on the sensitivity analysis result. As you mentioned, although Ethiopia has a great influence on the pooled prevalence, in fact, Madagascar has lowest prevalence and weight % is not so much different from Ethiopia but result does not change in sensitivity analysis while prevalence becomes doubled after omitting Ethiopia.

Author’s response: thank you for your constructive comment. According to your suggestion we omitted the sensitivity analysis in the revised manuscript and agreed to continue through table2 (see the revised version of this manuscript).

Reviewer 1: Figure 2 is not eye catching and the number are overlapped. Is the x-axis percentage? Better redraw it.

Author’s response: thank you for your productive comment. According to your suggestion we omitted the sensitivity analysis in the revised manuscript and agreed to continue through table2 (see the revised version of this manuscript).

Reviewer 1: LINE 218-236 – remove the subtitles and explanation of Table 2 results should go under the subtitle ‘Model comparison’. 

Author’s response: We’ve made this correction.

Reviewer 1: It is interesting that type of toilet is not associated with diarrhea. Is there any open defecation culture and such kinds of variables included in the dataset? If yes, you should consider it. Are every household have a toilet? 

Author’s response: no additional variable is added to this study beyond the mentioned variables in the manuscript. 

Reviewer 1: As Figure 2 and Table 3 give same information, omit one. 

Author’s response: We have removed figure 2 and decided to continue table 3 

Reviewer 1: Title for Table 3 is missing. A

Author’s response: thank you for your comment. The title of table 3 is added according to you comment (see the revised version of this manuscript).

Reviewer 1: LINE 249-259 – can omit as there is no information there.

Author’s response: we made this correction.

Reviewer 1: The authors tend to cover all information in the text and again in the table. You should only include the most important details in the text when you also have a table. 

Author’s response: Thank you for your suggestion, we have amended the comment accordingly.

Reviewer 1: The discussion is weak in light of the findings. The discussion section should be rewritten. The ideas are incoherently mixed. The discussion needs to focus on the key implications of the data with a separate paragraph for each concept.

Author’s response: some correction were made.

Reviewer 1: There are repetition of results in the discussion.

Author’s response: We’ve made this correction.

Reviewer 1: First paragraph of discussion, should start with brief answer to your research question. Do you think the diarrhea prevalence in East Africa is low or high? And why do you think? Is there any target? Etc.

Author’s response: We’ve made this correction.

Reviewer 1: In my perspective, comparing the pooled prevalence of East Africa and the prevalence of e.g. Ethiopia is not meaningful because Ethiopia is already included in the pooled prevalence. Furthermore, you mentioned the reason as ‘dissimilarity’ between East Africa and e.g. Ethiopia also not meaningful. 

Author’s response: We’ve made this correction

Reviewer 1: LINE 316-317 is confusing. Children have to provide supplementary food when the time arrives and it cannot be concluded as ‘highly likely to ingest unhygienic foods’. It mainly depends on the caregivers’ knowledge.

Author’s response: we have made this correction.

Reviewer 1: The defecation culture in East Africa (?open defecation), any efforts in WASH program, government’s implementation, caregiver’s knowledge, etc. should be discussed. 

Author’s response: we have made correction for this comment.

---

## [Decision Letter · Decision Letter 1]

25 Jan 2022

PONE-D-21-28429R1Pooled prevalence and associated factors of diarrhea among under-five years children in East Africa: A multilevel logistic regression analysisPLOS ONE

Dear Dr. Tareke,

Thank you for submitting your manuscript to PLOS ONE. After careful consideration, we feel that it has merit but does not fully meet PLOS ONE’s publication criteria as it currently stands. Therefore, we invite you to submit a revised version of the manuscript that addresses the points raised during the review process.

 Please submit your revised manuscript by Mar 11 2022 11:59PM. If you will need more time than this to complete your revisions, please reply to this message or contact the journal office at plosone@plos.org. Please include the following items when submitting your revised manuscript:A rebuttal letter that responds to each point raised by the academic editor and reviewer(s). You should upload this letter as a separate file labeled 'Response to Reviewers'.A marked-up copy of your manuscript that highlights changes made to the original version. You should upload this as a separate file labeled 'Revised Manuscript with Track Changes'.An unmarked version of your revised paper without tracked changes. You should upload this as a separate file labeled 'Manuscript'.

We look forward to receiving your revised manuscript.

Kind regards,

Khin Thet Wai, MBBS, MPH, MA

Academic Editor

PLOS ONE

Additional Editor Comments (if provided):

To consider and revise in line with reviewer's comments which are critical to improve scientific integrity.

Reviewers' comments:

Reviewer's Responses to Questions

**Comments to the Author**

1. If the authors have adequately addressed your comments raised in a previous round of review and you feel that this manuscript is now acceptable for publication, you may indicate that here to bypass the “Comments to the Author” section, enter your conflict of interest statement in the “Confidential to Editor” section, and submit your "Accept" recommendation.

Reviewer #1: (No Response)

Reviewer #2: All comments have been addressed

2. Is the manuscript technically sound, and do the data support the conclusions?

Reviewer #1: Partly

Reviewer #2: Yes

3. Has the statistical analysis been performed appropriately and rigorously? 

Reviewer #1: Yes

Reviewer #2: Yes

4. Have the authors made all data underlying the findings in their manuscript fully available?

Reviewer #1: Yes

Reviewer #2: Yes

5. Is the manuscript presented in an intelligible fashion and written in standard English?

Reviewer #1: No

Reviewer #2: Yes

6. Review Comments to the Author

Reviewer #1: PONE-D-21-28429R1

Pooled prevalence and associated factors of diarrhea among under-five years children in East Africa: A multilevel logistic regression analysis

Thank you for the revised version. Although there is some improvement, the manuscript still requires a significant improvement from the previous version of the manuscript. My sense is that the authors could not sufficiently explain or solve the reviewer’s comments and its current form does not have sufficient quality to warrant publication in PLOS ONE.

Comments

- Point by point responses with LINE number of corrected sentences would facilitate to review

- Attaching the old version of manuscript makes me confused

- English language correction is deemed necessary.

- Introduction should be clear, concise, and identify the added scientific value from the study.

- Table 2 – What is the difference between variables ‘mother’s current working status’ and ‘mother’s working status’. The proportion are different.

- You cannot control both ‘maternal education status’ and ‘community level educational status’ together in a regression model. According to your methodology, these two are derived from a single variable/information, correct? Same applied to poverty/wealth index.

- Also, did you check collinearity?

- Regarding relation between diarrhea and time of breast feeding initiation, I also do not think these are related. That’s the reason why I am asking to you why because you study population is under 5 children and you controlled breastfeeding initiation in the regression model.

- Discussion section needs to be improved a lot.

Reviewer #2: (No Response)

7. PLOS authors have the option to publish the peer review history of their article (what does this mean?). If published, this will include your full peer review and any attached files.

Reviewer #1: No

Reviewer #2: No

---

## [Author Response · Author response to Decision Letter 1]

3 Feb 2022

Author’s response to reviews

Title: Pooled prevalence and associated factors of diarrhea among under-five years children in East Africa: A multilevel logistic regression analysis

Authors:

Abiyu Abadi Tareke (abiyu20010@gmail.com)

 Ermias Bekele Enyew (Ermiashi@gmail.com)

Bayley Adane Takele (behaileadane@gmail.com)

Version: 2

 Date: 3 February 2022

Author’s response to reviews: 

Plos one 

Point by point response for editors/reviewers comments

Manuscript title: Pooled prevalence and associated factors of diarrhea among under-five years children in East Africa: A multilevel logistic regression analysis

Manuscript ID: PONE-D-21-28429

Dear editors/reviewers:

Dear all,

We would like to thank you for your generous, revealing and constructive comments to the second version of the manuscript. You valuable comments would advance the substance and content of the manuscript. The authors accounted each comments and clarification questions of editors and reviewers in a focused way.

Our point-by-point responses to each comment and questions are detailed on the following pages. Further, the details of changes were shown by track changes in the supplementary document attached.

Comments

- Point by point responses with LINE number of corrected sentences would facilitate to review

- Attaching the old version of manuscript makes me confused

- Author’s response: We made correction to this comments 

- English language correction is deemed necessary.

- Author’s response: We made some correction to this comment

- Introduction should be clear, concise, and identify the added scientific value from the study.

- Author’s response: Some scientific values are added

- Table 2 – What is the difference between variables ‘mother’s current working status’ and ‘mother’s working status’. The proportion are different.

- Author’s response: Thank you reviewer for your constructive comment. We accept and modified it. The variable named “mother’s current working status” was added to the table mistakenly. The difference in the proportion is occurred because of this variable was tabulated without adding the command of sample weighting. That way it has lower frequency compared to the second variable ‘mother’s working status’. As the variable ‘mother’s working status’ has weighted sample we decided to continue with this variable and we removed the variable having un-weighted frequency. 

- You cannot control both ‘maternal education status’ and ‘community level educational status’ together in a regression model. According to your methodology, these two are derived from a single variable/information, correct? Same applied to poverty/wealth index.

- Author’s response: we thank you for your valuable comment. Community level factors are pertinent factors for children health services, we considered the community level factors by aggregating them from the respective individual level factors to indicate the neighboring effect. This helps policymakers to take intervention at both individual and community levels. For example, child born from mother who is from communities with lower level of community education might be clustered in specific location and taking appropriate intervention in this group of women could have a great advantage to increase child health services including diarrhea prevention and treatment services. In our opinion educational status at individual level are different from educational status generated at community level. 

- Also, did you check collinearity?

- Author’s response: Thank you reviewer for your concern. Yes, we have checked it. The multicollinearity issue was addressed using pseudo linear regression analysis through applying the command “estat vif” and the mean variance inflation factor (VIF) of 2.31. Which ensure that non –presence of multicollinearity b/n predictor variables.

- Regarding relation between diarrhea and time of breast feeding initiation, I also do not think these are related. That’s the reason why I am asking to you why because you study population is under 5 children and you controlled breastfeeding initiation in the regression model.

- Author’s response: Time of breastfeeding initiation is included in this study because of early initiation of breast feeding has many health benefit. According to WHO feeding colostrum in the first hour increases the likelihood babies will continue to be breastfed which gives them a head start in the “race against malnutrition“. This means children who had history of early breast feeding initiation are less likely to encounter malnutrition and if the probability of encountering malnutrition is low the probability of diarrhea will also decrease. Additionally, the more initiation of breast feeding early the more likely to continue their breast feeding properly. And this will reduce the frequency of diarrhea. Therefore, we included this variable considering those scientific backgrounds.

- Discussion section needs to be improved a lot.

- Author’s response: Thank you a lot for your comment. We accept all your comments and modified it accordingly.

---

## [Decision Letter · Decision Letter 2]

14 Feb 2022

Pooled prevalence and associated factors of diarrhea among under-five years children in East Africa: A multilevel logistic regression analysis

PONE-D-21-28429R2

Dear Dr. Tareke,

We’re pleased to inform you that your manuscript has been judged scientifically suitable for publication and will be formally accepted for publication once it meets all outstanding technical requirements.

Kind regards,

Khin Thet Wai, MBBS, MPH, MA

Academic Editor

PLOS ONE

Additional Editor Comments (optional):

All comments are adequately addressed.

Reviewers' comments:

Reviewer's Responses to Questions

**Comments to the Author**

1. If the authors have adequately addressed your comments raised in a previous round of review and you feel that this manuscript is now acceptable for publication, you may indicate that here to bypass the “Comments to the Author” section, enter your conflict of interest statement in the “Confidential to Editor” section, and submit your "Accept" recommendation.

Reviewer #1: All comments have been addressed

2. Is the manuscript technically sound, and do the data support the conclusions?

Reviewer #1: Yes

3. Has the statistical analysis been performed appropriately and rigorously? 

Reviewer #1: Yes

4. Have the authors made all data underlying the findings in their manuscript fully available?

Reviewer #1: Yes

5. Is the manuscript presented in an intelligible fashion and written in standard English?

Reviewer #1: Yes

7. PLOS authors have the option to publish the peer review history of their article (what does this mean?). If published, this will include your full peer review and any attached files.

Reviewer #1: No

---

## [Editor Report · Acceptance letter]

6 Apr 2022

PONE-D-21-28429R2 

Pooled prevalence and associated factors of diarrhea among under-five years children in East Africa: A multilevel logistic regression analysis 

Dear Dr. Tareke:

I'm pleased to inform you that your manuscript has been deemed suitable for publication in PLOS ONE. Congratulations! Your manuscript is now with our production department. 

Kind regards, 

on behalf of

Dr. Khin Thet Wai 

Academic Editor

PLOS ONE